

# The role of geomorphic zonation in long-term changes in coral-community structure on a Caribbean fringing reef

Alexis Enrique Medina-Valmaseda[1,2], Rosa E. Rodríguez-Martínez[2], Lorenzo Alvarez-Filip[2], Eric Jordan-Dahlgren[2] and Paul Blanchon[2]

[1] Posgrado en Ciencias del Mar y Limnología, Universidad Nacional Autónoma de México, Coyoacán, Ciudad de México, Mexico
[2] Instituto de Ciencias del Mar y Limnología, Universidad Nacional Autónoma de México, Puerto Morelos, Quintana Roo, Mexico

## ABSTRACT

Ecological processes on coral reefs commonly have limited spatial and temporal scales and may not be recorded in their long-term geological history. The widespread degradation of Caribbean coral reefs over the last 40 years therefore provides an opportunity to assess the impact of more significant ecological changes on the geological and geomorphic structure of reefs. Here, we document the changing ecology of communities in a coral reef seascape within the context of its geomorphic zonation. By comparing basic ecological indices between historical and modern data we show that in 35 years the reef-front zone was transformed from a complex coral assemblage with a three-dimensional structure, to a size-homogenized and flattened one that is quasi indistinguishable from the adjacent non-accretional coral-ground zone. Today coral assemblages at Punta Maroma are characterized by the dominance of opportunistic species which are either tolerant to adverse environmental conditions, including sedimentation, or are known to be the first scleractinian species to recruit on disturbed reefs, implying they reflect a post-hurricane stage of adjustment. Despite an increase in similarity in ecological indices, the reef-front and coral-ground geomorphic zones still retain significant differences in coral assemblages and benthic habitat and are not homogeneous. The partial convergence of coral assemblages certainly has important consequences for the ecology and geological viability of the reef and its role in coastal protection, but environmental physical drivers continue to exert a fundamental role in the character and zonation of benthic communities of this reef seascape.

# INTRODUCTION

Coral reefs develop over geologic timescales through a complex process termed accretion (*Perry & Hepburn, 2008*; *Pandolfi et al., 2011*). Most consider this to involve a dynamic interplay between three ecological and sedimentological processes: framework growth, physical and biological erosion and internal sedimentation and cementation (*Rasser & Riegl, 2002*; *Perry & Hepburn, 2008*). Framework growth is accomplished primarily by

Corresponding author
Alexis Enrique Medina-Valmaseda,
alexismedina67@gmail.com

scleractinian corals but with important contributions from calcifying encrusters, such as crustose coralline algae (*Adey, 1975*; *Kikuchi & Leão, 1997*). This growth is balanced by biological, chemical and physical erosion and mediated by environmental gradients in wave-energy, light penetration, and sediment flux (*Geister, 1977*; *Huston, 1985*), and produces a layer of geomorphically zoned framework (*Geister, 1977*; *Graus & Macintyre, 1989*; *Kennedy & Woodroffe, 2002*). Left undisturbed by storms and hurricanes, this framework has the potential to accrete vertically, as one coralgal cohort develops over another through time producing a geological reef deposit (*Geister, 1980*; *Done, 2011*).

Vertical reef accretion is clearly dependent upon short-term ecological processes persisting over thousands of years and generating a positive balance of growth and accumulation over erosion and removal. And even in zones with a positive balance, that accretion should vary significantly depending on the size and growth rate of corals in assemblages dominating each zone. For example, analysis of cores drilled on Caribbean reefs protected from hurricanes has shown that frameworks dominated by branching Acroporid corals have undergone significant vertical accretion and produced large geological reef structures during the Holocene (*Macintyre & Glynn, 1976*). However, in more hurricane-prone areas, cores obtained from the same zones have shown that Holocene reef structures are biodetrital consisting mostly of the fragmented remains of corals that once covered their surfaces (*Blanchon, Jones & Kalbfleisch, 1997*; *Blanchon et al., 2017*). Furthermore, non-accretionary coral-grounds, have been reported from the extensive shelf zones both adjacent to and between the accretionary reef structures (*Rodríguez-Martínez et al., 2011*). Clearly the ecological seascape of reefs is complex and transient and surface coral assemblages may not always indicate how accretion occurs or even if it occurs at all.

With the demise and deterioration of Caribbean reefs during the last 40 years, and the decimation of keystone Acroporids in particular (*Gardner et al., 2005*; *Jackson et al., 2014*), this complex ecological seascape is radically changing (*Perry & Alvarez-Filip, 2019*). Biodiversity loss and biotic homogenization are not only compromising the ecological functioning of reefs (*Olden & Poff, 2003*; *Burman, Aronson & Van Woesik, 2012*; *Alvarez-Filip et al., 2013*; *Elliff & Silva, 2017*) and their ability to provide local and regional services (*Alvarez-Filip et al., 2009*, *2011*), but are predicted to reduce their potential for long-term accretion (*Perry et al., 2013*; *Estrada-Saldívar et al., 2019*). Assessing the accuracy and validity of these accretion predictions, however, is problematic for several reasons (*Lange, Perry & Alvarez-Filip, 2020*). First, biotic homogenization stemming from the loss of large reef-builders, like Acroporids, makes it difficult to locate geomorphic boundaries between accretionary fringing-reef zones and adjacent non-accretionary coral-ground zones that veneer the surrounding shelf. Combining such geomorphic zones would therefore give less representative accretion potentials for reef systems. Second, the use of ecological "snapshots" to estimate the accretion potential of entire reefs makes two questionable assumptions: that drivers which exist outside of ecological timescales are unimportant in the accretion process, and that accretion is uniform in space and time (*Perry, 2011*; *Courtney et al., 2016*; *Manzello et al., 2018*). In terms of drivers, we know that hurricanes have played an important role in Caribbean reef accretion during the Holocene

(*Blanchon et al., 2017*). However, little is known about changes in accretion rates, although it seems unlikely that they would remain constant given that sea level and climate has varied during the Holocene (*Blanchon & Shaw, 1995*; *Gischler, 2006*; *Toscano & Macintyre, 2006*; *Khan et al., 2017*).

In this study, we assess if homogenization has occurred between two most adjacent windward geomorphic zones along a reef at Punta Maroma, Mexico, where the long-term history of accretion is known from drilling (*Blanchon et al., 2017*). These geological records show the geomorphic zones have different accretion histories, and yet ecological studies commonly group them into a general "fore-reef" zone. Although our analysis of historical vs recent ecological surveys from these zones indicates that there has been some homogenization in the abundance, species composition, and size structure of coral assemblages in adjacent geomorphic zones, we find that coral communities continue to be statistically different.

## METHODS

### Study site

The study site is a 4.5 km-long fringing reef at Punta Maroma, in the northeastern Yucatan Peninsula, close to Playa del Carmen, Quintana Roo, Mexico (Fig. 1A). It has a typical tripartite geomorphic zonation, with a reef-front, crest, and back-reef, and is flanked by a shallow (<6 m) lagoon on its landward side, and a deeper (>8 m) coral-veneered rock terrace, on its seaward side (Fig. 1B). The geological structure of the reef front was reported by *Blanchon et al. (2017)* who showed that it consists of clast-dominated hurricane deposits, with a maximum age of 5.5 ka. Cores from this study also showed that the seaward coral-veneered rock terrace is a late Pleistocene extension of the coastal bedrock with no evidence of reef accretion during the Holocene.

The combination of these geomorphic zones produces an extensive windward ecological seascape (>2,000 m$^2$), which consists of a shallow accretionary reef-front zone (RF), from the crest down to ~6 m depth (Fig. 1C), and a non-accretionary deeper coral-ground (CG) zone (Fig. 1D), extending from the limit of the RF out across the rock-terrace to a mid-shelf slope break (Fig. 1E) at ~10 m depth (*Rodríguez-Martínez et al., 2011*; *Blanchon et al., 2017*). These zones have been collectively referred to as the "fore-reef" in other studies (Fig. 1B).

### Survey methodology and historical analysis

To compare recent coral assemblages from the RF and CG zones at Punta Maroma reef, we surveyed coral species abundance and size between March and June, 2019. The sampling effort consisted of ten 30 m-long belt transects (BT), randomly placed in an orientation roughly parallel to the crestline at ~5 m (RF) and ~10 m (CG). Sample size was determined based on historical cumulative species diversity curves, empirically determined in previous studies (*Jordan et al., 1981*; *Jordán-Dahlgren, 1989*). All scleractinian colonies with size ≥4 cm within 1-m-wide belt transects were sampled, including those intercepted by the belt line, following *Zvuloni et al. (2008)*. Other environmental data, including depth, spatial position and distance to the mid-shelf edge,

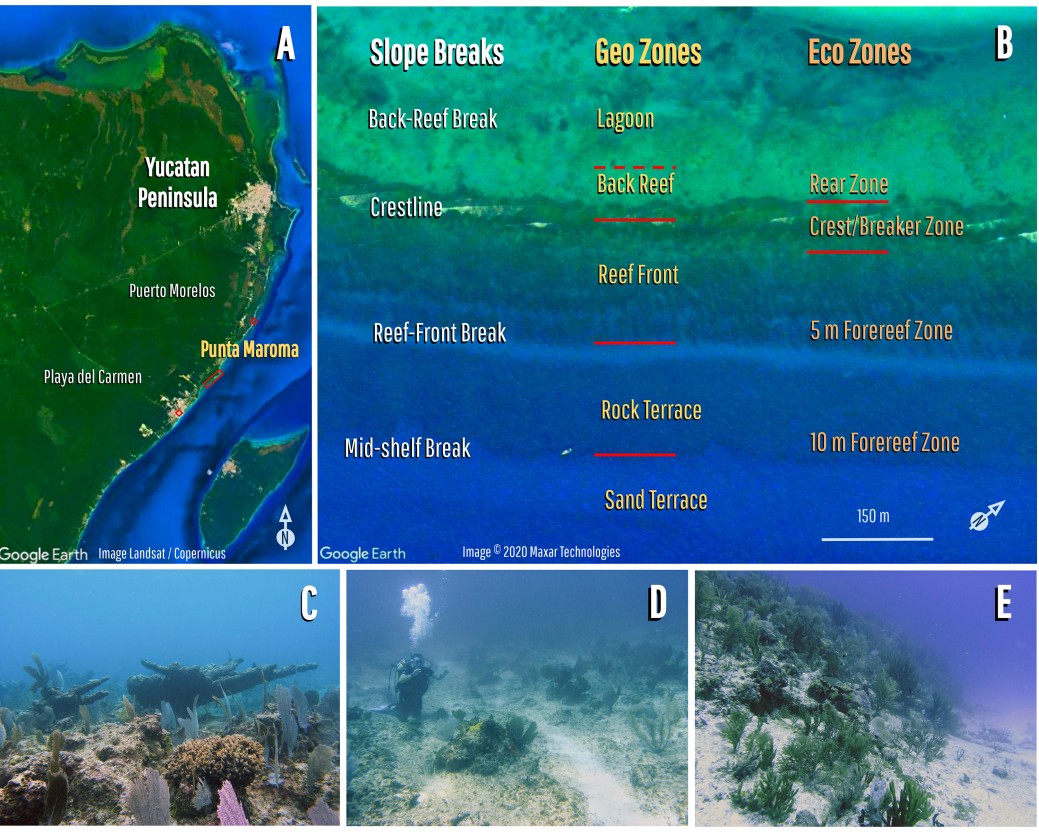

**Figure 1 Ecological seascape and zonation of the fringing reef at Punta Maroma.** (A) Location of Punta Maroma; (B) reef zonation and geomorphology showing: slope breaks and geomorphological zones (following *Blanchon et al., 2017*) and ecological seascape zones (following *Jordan et al., 1981*; *Estrada-Saldívar et al., 2019*); (C) view of the reef-front zone (or fore reef) at ~5 m; (D) view of the rock-terrace zone (or fore reef) at ~10 m; (E) view of the mid-shelf break (or fore reef) between 10 and 15 m.

were also recorded. Scleractinian coral species were classified based on their morphology and life history following *Estrada-Saldívar et al. (2019)*. We informed the authorities of Comision Nacional de Areas Naturales Protegidas (CONAMP) prior to conducting fieldwork as it is a requirement in order to carry out activities within MPA.

To determine if coral assemblages from the RF and CG zones at Punta Maroma reef experienced changes in the last few decades, we used ecological surveys taken in 1979 (*Jordan et al., 1981*) and in 1985 (*Jordán-Dahlgren, 1989*). These data were obtained from 20-m long line-intercept transects (LIT), separated from each other by distances of 5–25 m. In these studies, the transects were delimited by plastic chains (with 2.73 cm chain links) that followed the bottom topography. Transects were taken from three zones, described as "rear-reef", "breaker zone" and "fore-reef". In both surveys the "fore-reef" transects were placed perpendicular to the coast at two depths: 5 and 10 m. These three zones correspond to four geomorphological zones described by *Blanchon et al. (2017)*: the Back Reef ("rear reef"), Reef Crest ("breaker zone"), Reef Front ("forereef" at 5 m) and Rock Terrace and Mid-shelf break ("forereef" at 10 m). In this study we refer to the "forereef" at 10 m as the coral-ground. All scleractinian corals below the chain were

counted and measured using the chain link as the measurement unit. In 1979, the number of transects per depth was five and in 1985 four. Additionally, in 1985 colonies were measured by their maximum diameter. Every survey fulfilled the minimal sampling effort necessary to accurately describe the coral communities according to cumulative species diversity curves (*Gleason, 1922*). Also, because in the 1979 and 1985 *Montastrea annularis, Montastraea faveolata* and *M. franksi* were considered as part of the same species complex (*M. annularis*), for our 2019 surveys we combined these three species in one (*Orbicella* (formerly *Montastrea*) "*annularis*" complex).

## Approach to accretion processes

Scleractinian corals are considered to be the principal builders of three-dimensional structures of tropical coral reefs (*Goreau, 1959*). However, differences in skeleton stability, size and habitat distribution of coral species, can lead to differences in their preservation potential in the geological record (*Greenstein, 2007*). Therefore, we use a two-level classification of the role of coral species in the accretion process to reflect the presence or species contribution to the geological records. Levels were: (a) key reef-building species (key spp.), consisting of large branching Acroporids and massive *Orbicella* (formerly *Montastrea*) "*annularis*" complex (*Budd et al., 2012*), which are considered to be the principal reef builders in the Caribbean (*Goreau, 1959*; *Lewis, 1984*; *Toth et al., 2019*) and (b) less influential species like small massive, sub-massive or encrusting, digitate and foliose morphologies with lower growth rates, which participate in the accretion processes but are less represented in the geological records. This group includes some abundant species in the contemporary coral communities like *Agaricia agaricites* and *Porites astreoides* (*Aronson, 2006*; *González-Barrios & Álvarez Filip, 2018*; *Toth et al., 2019*).

In addition to classifying their role in accretion, we also use a species Importance Value Index (IVI) (*Curtis & McIntosh, 1951*; *Finol Urdaneta, 1971*) as a proxy to estimate the relative importance of each species in the accretion processes within each geomorphic zone. For this estimation in the pre-1990s period, we used only 1985 dataset since information of 1979 data was incomplete. The IVI of each species is calculated as IVI = (RA + RSD + RF)/3, where RA is relative abundance, calculated from the number of individuals per species with respect to the number of individuals of all species found in the community; where RSD is relative spatial dominance defined as the area covered by each species (using the colonies maximum and minimum diameters and assuming a planar area for the 2019 data) with respect to the cover of all species; and where RF is relative frequency, estimated as the proportion of transects where a species is present, normalized to the frequency of all species in the community. The analysis of IVI was carried out because different geomorphic zones within a reef have a heterogeneous accretion capacity as a consequence of the composition of the coral community and external environmental gradients (*Geister, 1977*; *Perry, 1999*).

## Method biases and uncertainties

Different benthic sampling protocols and probabilistic designs tend to produce somewhat different results, and therefore different measures and indices with their own particular

bias (*Vallès, Oxenford & Henderson (2019)* and others). However, if methodological bias is small relative to the real signal of change, then the latter should be detectable. Benthic survey methods are characterized by different spatial grain sizes of sampling units, and therefore the resulting abundance measurements could be described by different sampling distributions. However, *Nadon & Stirling (2006)* and *Jokiel et al. (2015)* reported that the method-biases associated to the results from different benthic protocols (including the two used to collect the data we analyze here), are small relative to the true signal of changes. By contrast, *Vallès, Oxenford & Henderson (2019)* recently claimed that switching between benthic methods and pooling data without accounting for method differences could obscure real tendencies, for example in coral coverage percentages. The actual impact of this important source of error in further analyses of data collected by distinct benthic methods depends on the specific question to be answered and how data are treated (*Ohlhorst et al., 1988*; *Chiappone & Sullivan, 1991*; *Beenaerts & Berghe, 2005*). For instance, in our study, the comparative analysis was focused mostly in spatial and temporal changes in diversity composition and the correlation between species proportions by geomorphic zones, rather than in the estimation of temporal tendencies in any particular ecological index (e.g., coral cover). Yet, transect methods yield roughly the same estimates for coral genus diversity but tend to overestimate coral densities and coral coverage (*Ohlhorst et al., 1988*; *Beenaerts & Berghe, 2005*). Therefore, we consider our analysis to be more affected by an incomplete representation of the community composition rather than another method bias. We tackle the uncertainty related to the accurate representation of coral species within each geomorphic zone reaching the asymptotic zone of species accumulation curves in each zone. Ultimately, we are aware that ecological data, including abundance data, usually violate the assumptions of traditional parametric statistics, which makes it difficult to deal with using traditional univariate methods (*Fieberg, Vitense & Johnson, 2020*).

## Data transformation and statistical analyses

To tackle the challenge of statistical analysis of empirical ecological data, we selected a methodological approach based on multivariate analysis of abundance data. Although having some limitations, a multivariate approach has been recognized as more effective in dealing with ecological data (*Beals, 1984*; *Fieberg, Vitense & Johnson, 2020*). The approach includes several standard transformations of data, so the raw data were not compared. First, the absolute abundance data from the 2019 survey were transformed to relative spatial living coverage estimated from average colony maximum diameter and the number of individuals. This was then standardized by its total to reduce the differences in magnitude generated by the benthic method. Although the abundance data from 1979 and 1985 were relative measurements (percentage of live coral coverage), we standardized values again by its total to reduce method-bias related to the effect of the former benthic scheme that included other benthic classes (*Vallès, Oxenford & Henderson, 2019*). In addition, data were square-root transformed to reduce the differential between the largest and smallest non-zero value in the transformed matrix. The relativization process (percentage of the total sample) was followed by an ordination process where we

constructing the correspondent Bray–Curtis matrix of similarities. Additionally, each data set was transformed into Presence/Absence data and generated correspondent Jaccard matrix. Each specific multivariate test demands its additional pre-treatment procedure for the best performance. All subsequent multivariate statistical analyses were performed using Plymouth Routines in Multivariate Ecological Research (Primer-e version 7.0.13, serial number 4901) (*Clarke & Gorley, 2015*). Graphical outputs were constructed using the free software platform of *R Core Team (2013)* and Primer-e v7.0.13.

To determine the contribution of species to coral assemblages in each zone through time, we conducted a two-way similarity percentage analysis (SIMPER) for zones by time period, based on Bray–Curtis similarity measures of transformed square-root matrix of abundance data, making a 70% cut-off for low contributions (*Clarke & Warwick, 1994*; *Clarke et al., 2014*). To evaluate changes in diversity we conducted a PERMANOVA and PERMDISP tests on the basis of a Jaccard transformed data matrix using previously transformed to Presence/Absence data. PERMDISP test on the basis of Jaccard matrix has been reported as a reliable test to identify changes in beta-diversity (*Anderson, 2006*; *Anderson, Ellingsen & McArdle, 2006*). Shade plots were created to visualize the relative contribution of all surveyed coral species to the assemblages of each geomorphic zone before the 1990s and in 2019.

To compare coral assemblages between geomorphic zones we performed two pairwise PERMANOVA tests for zones by time period, based on Bray–Curtis similarity measures of transformed square-root matrix of abundance data and in Jaccard matrix of presence-absence. To evaluate changes in coral communities of the two zones through time, we performed a two-way crossed permutational analysis (PERMANOVA) of the Bray–Curtis matrix under an orthogonal design of two fixed factors: time, with three levels (1979, 1985 and 2019), and zone, with two levels (RF and CG). Additionally, for the factor year, we designated two linear contrasts: C1 (1979 vs 1985) and C2 (1979 and 1985 vs 2019). Linear contrast is a statistical procedure that allows comparing different sub-sets of data, whereas other sub-sets can be excluded from the comparison. Search for the possibility of pooling the data subsets from 1979 and 1985 stemmed from the previous knowledge that the 1990s decade is considered a turnover moment for coral reefs worldwide, including Punta Maroma reef (*Odériz et al., 2014*). Therefore, we expected no differences between those data whereas the pooling process leads to a balanced design in further analysis of periods of interest. The test was done using permutation of residuals under a reduced model and Type III (partial Square Sums) in 9,999 permutations (*Anderson, 2001*, *2017*; *Anderson & Braak, 2003*). To measure and test the homogeneity of multivariate dispersions of data, we performed a non-parametric permutational analysis of multivariate dispersions (PERMDISP), along with pairwise comparisons of Bray–Curtis matrix of similarities. PERMDISP was performed on the basis of distances to centroids, with $P$-values obtained using permutations ($P$ (perm)) and 9,999 permutations, giving the best overall results expected in terms of type I error and power (*Anderson, 2006*). The results are presented in a Principal coordinates analysis (PCO; (*Torgerson, 1958*; *Gower, 1966*)) constructed by calculating the distances between samples in a transformed Bray–Curtis similarity merged matrix of previously standardized

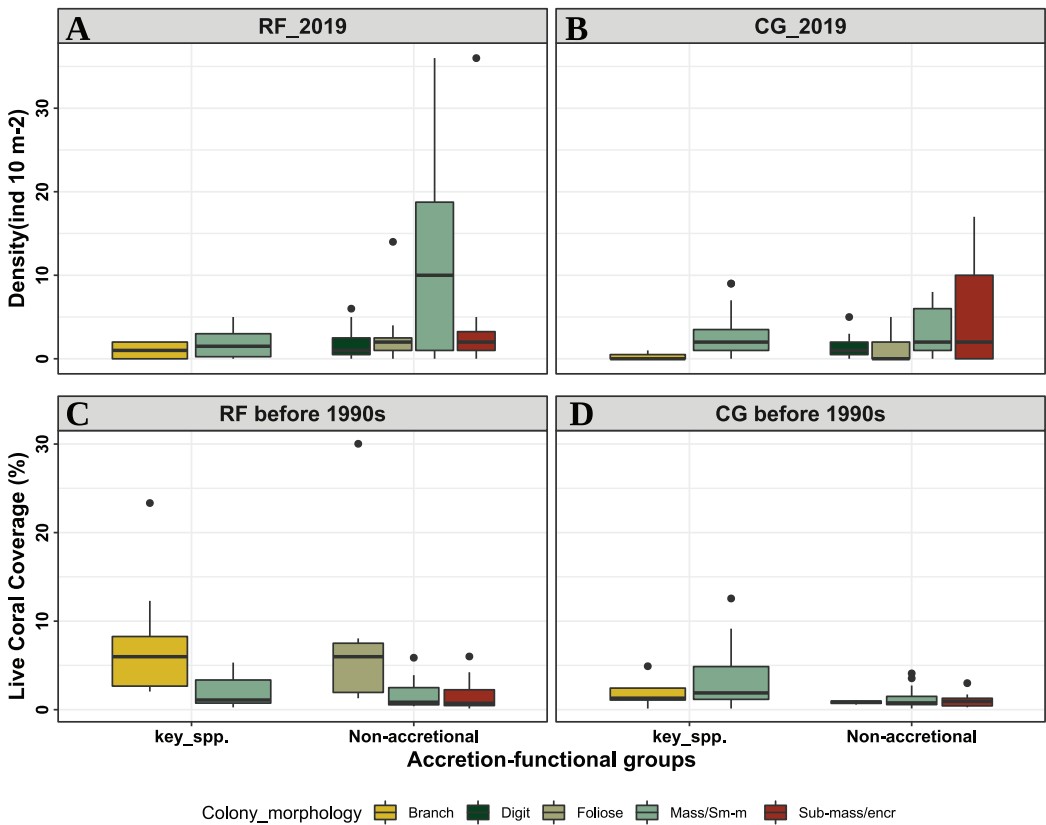

**Figure 2 Box Plots of Coral colony densities in reef-front (A) and coral-ground (B) zones in 2019.**
Box Plots (C) and (D) show Live coral coverage in reef front and coral-ground zones respectively before the 1990s, by accretion-functional groups, coral morphology and geomorphic zone in two reef zones at Punta Maroma reef. For the period before 1990s (C and D), data from 1979 and 1985 were pooled together. The bottom and top of the box are the first and third quartiles, respectively, the black line inside the box is the median. Whiskers are the lowest datum still within 1.5 times that of the lower quartile and the highest datum still within 1.5 times that of the upper quartile. RF: Reef front, or accretionary zone, CG: Coral-ground or non-accretionary zone; key spp.: key "reef-building" species; non-accretional: less influential species for accretion processes.

and square-root transformed relative abundances matrices, for both periods in time. Pearson correlation coefficients of selected taxa were superimposed over the PCO as vectors to indicate the taxa that most strongly contributed to reef community variation. The taxa selected were derived from a similarity percentage breakdown (SIMPER) analysis of the characteristic and distinguishing reef taxa.

## RESULTS

### Contemporary assemblages

In 2019, we identified and measured 724 coral colonies in the RF zone and 732 colonies in the CG zone, resulting in coral densities of 2.41 0.89 and 2.44 0.95 col·m$^{-2}$ respectively (Figs. 2A and 2B). In the RF zone, coral colonies belong to 23 species of 13 genera, and five species account for 89.4% of the colonies in the sample: *Porites astreoides* (55.7%), *Agaricia agaricites* (11.9%), *Siderastrea siderea* (9.4%), *Agaricia tenuifolia* (6.9%), and

**Table 1 Coral species and number of colonies recorded in two geomorphic zones at Punta Maroma reef seascape before 1990s and in 2019.** Coral species are classified according to their growth morphology (Darling et al., 2012). CG: Coral-ground zone, RF: Reef front zone of the fringing reef. Key reef-building spp.: large "reef-building" species, Non-accretional Spp.: less influential species for accretion processes.

| Accretion potential of species and Colony morphology | Species richness | Before 1990s | | Species richness | 2019 | |
|---|---|---|---|---|---|---|
| | | RF | CG | | RF | CG |
| **Key reef-building Spp.** | | | | | | |
| **Massive:** | 4 | 11 | 26 | 7 | 78 | 253 |
| *Colpophyllia natans, Diploria labyrinthiformis, Montastraea cavernosa, M. annularis spp. Complex (Orbicella faveolata, O. annularis), Pseudodiploria. strigosa, Siderastrea siderea.* | | | | | | |
| **Branching:** | 2 | 16 | 5 | 3 | 15 | 5 |
| *Acropora palmata, A. cervicornis, A. prolifera* | | | | | | |
| **Non-accretional Spp.** | | | | | | |
| **Small massive:** | 9 | 10 | 26 | 6 | 408 | 136 |
| *Solenastrea bournoni, Isophyllia rigida, Favia fragum, Dichocoenia stokesii, Meandrina meandrites, Porites astreoides* | | | | | | |
| **Sub-massive or encrusting:** | 3 | 11 | 14 | 3 | 121 | 254 |
| *Agaricia agaricites, S. radians, Stephanocoenia intersepta* | | | | | | |
| **Digitates:** | 2 | 3 | 0 | 3 | 49 | 50 |
| *Porites porites, P. furcata, P. divaricata* | | | | | | |
| **Foliaceous:** | 1 | 7 | 0 | 5 | 53 | 34 |
| *Agaricia fragilis, A. humilis, A. tenuifolia, A. lamarcki, Helioseris cucullata, Mycetophyllia lamarckiana* | | | | | | |
| **Total number of colonies by zone** | | 55 | 74 | | 724 | 732 |

*Porites porites* (5.5%); the other 18 species represent the remaining 10.6% of the sample, with none representing over 5% (Table 1). In the CG zone, coral colonies belong to 23 species of 16 genera, and five species account for 81.5% of the sample: *A. agaricites* (32.5%), *P. astreoides* (15.8%), *Montastraea cavernosa* (10.2%), *S. siderea* (16.4%), and *P. porites* (6.4%). The coral species that are considered to contribute most to reef accretion (Acroporids and some massive forms) represent 35% of colonies in the non-accretionary CG zone, whereas they represent only 12.8% in the RF zone with the species *Acropora palmata* represented by a minimal number of individuals. In both zones combined, these "key" reef builders represent 23.9% of all colonies.

Coral colony sizes in both the RF and CG zones are predominantly small, independent of their morphology (mean = 17.9 ± 14.7 cm in the RF and 19.3 ± 15.4 cm in the CG) (Data S1A and S1B), with only 2.5% of them having diameters larger than 50 cm in both zones. Additionally, coral colonies of all morphologies have low heights in both zones (mean = 6.5 ± 8.0 cm in the RF and 10.8 ± 11.1 cm in the CG). Corals with massive and sub-massive encrusting morphologies dominate both zones contributing 83.8% of the colonies in the RF and 87.8% in the CG (Fig. 2A), but the identity of the dominant species differs, with the small massive *P. astreoides* dominating in the RF zone and the sub-massive

encruster *A. agaricites* in the CG zone. The SIMPER test shows the groups of species that co-occur between transects (Data S2), and indicates that four species have a high degree of overlap within the CG zone: *A. agaricites, S. siderea, P. astreoides*, and *M. cavernosa* (Average similarity: 70.7), whereas in the RF zone three species overlap: *P. astreoides, A. agaricites* and *S. siderea* (Average similarity: 58.4).

## Historical assemblages

Historically, all coral morphologies had higher coral coverage in the RF zone than in the CG zone; the RF zone showed a dominance of branching morphologies and the CG zone a dominance of massive ones (Table 1; Figs. 2C and 2D). The SIMPER test for historical data shows three species overlapped in the RF zone: *Ac. palmata, Acropora cervicornis*, and *Ag. tenuifolia* (Average similarity: 52.8) and five in the CG zone: *M. cavernosa, Dichocoenia stokesii, S. siderea, Ag. agaricites*, and *Orbicella* complex (Average similarity: 42.6) (Data S1). In the RF zone, *Ag. tenuifolia* and *Ac. palmata* accounted for the largest live coral coverage (30.0% and 23.3% respectively), and in the CG zone the dominant species were *Orbicella* species and *M. cavernosa* (12.6% and 9.2% respectively, Table 1). In 1985, the average diameter maximum of coral colonies (mean = 42.1 ± 24.4 cm) was more than double of those in 2019 (mean = 18.6 ± 3.2 cm, Data S1A).

According to the Importance Value Index, the main species in the RF zone in 1985 were *P. astreoides, Ag. tenuifolia* and *Ac. palmata* and in 2019 they were *P. astreoides, Acropora prolifera* and *Ac. palmata*. The main species in the CG zone in 1985 were *Orbicella* species, *M. cavernosa, P. astreoides* and *Pseudodiploria clivosa*, and in 2019 they were *Ag. agaricites, Colphophilia natans* and *S. siderea*. So in RF zone, the relative importance of *Ac. palmata* and *P. astreoides* increased in time whereas that of *Ag. tenuifolia* decreased, and in CG zone the dominant species were replaced (Fig. 3; Data S3).

## Comparative analysis of coral communities

Coral assemblage of the CG zone underwent significant changes in beta diversity (PERMDISP, Jaccard matrix, Fc: 4.59, $P$ (perm) < 0.01; Table 2; Data S4A), as shown in the shade plot (Fig. 4). Changes in the coral community composition of the two zones and heterogeneity in species distribution were analyzed using a two-way crossed (orthogonal) PERMANOVA. This shows strong balanced effect of zone- and time-factors over coral assemblages ($P$ < 0.001 for each test; Estimates of components of variations: 1155.5 and 1105.5 respectively; Data S4B). The test indicates a statistically significant interaction in the effects of zone and time ($P$ < 0.001), although the combined effect was lower (Estimates of components of variations: 390.9). Linear contrasts indicate that the effect of time was relevant when comparing 2019 vs before 1990s although there was an effect of zonation. A posteriori PERMANOVA pairwise test for both zone- and time-factors vs the zone factor, shows that the average similarity between FR and CG groups is 52.6 (Table 2; Data S4B).

These differences in abundance and composition of coral assemblages for the two zones through time are mirrored in a Principal Coordinates Analysis (PCO) ordination procedures (Fig. 5; Data S5), which shows that *A. agaricites, S. siderea, M. cavernosa*, and

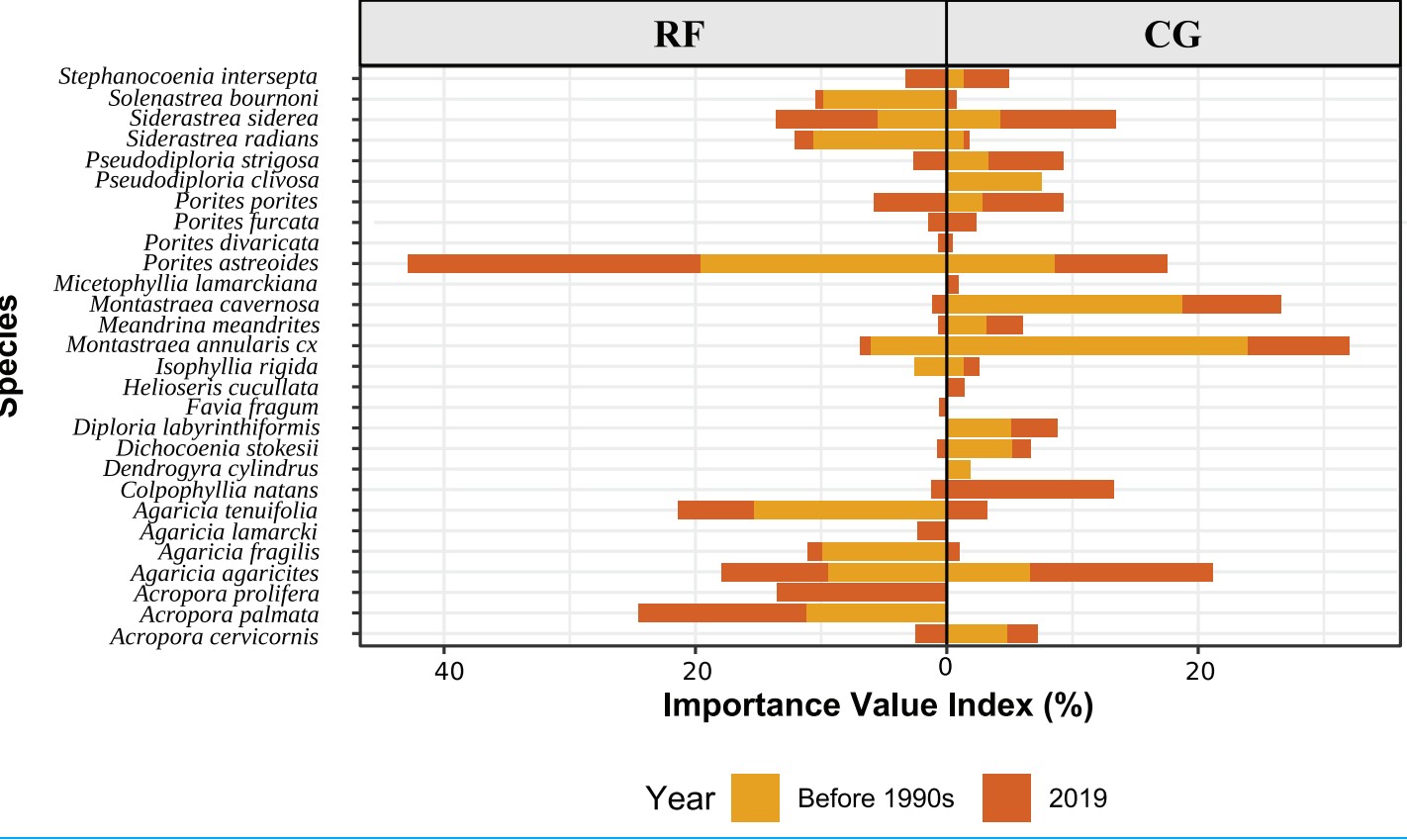

**Figure 3** Ecological Importance Value Index of coral species by geomorphic zones in Punta Maroma reef before 1990s (based on 1985 data) and in 2019. The index is based on the relative abundance, frequency and spatial contribution of each species with respect to the coral assemblage. The *x* axis represents the importance value index (IVI) in percentages and the *y* axis lists the coral species by it scientific accepted names. RF: Reef front, CG: coral-ground.

*P. astreoides* have a strong negative relationship with the PCO1 axis (indicative of 2019 sites), while *Ag. agaricites, S. siderea* and *Ac. cervicornis* are neutrally related to the PCO2 axis. However, the main reef builders (acroporids and orbicellids) are strongly and positively related to the PCO1 axis (indicative of the period before 1990s), with *A. palmata* being positively related to RF zone before 1990s, whereas species of *M. annularis* spp. complex are related to CG zone (Fig. 5). Other species, such as *M. cavernosa, D. stokesii,* and *P. strigosa*, have strong negative relationships with negative sections of both axes (indicative of the CG zone). The two main axes of PCO based on abundance data reflect 35.8% (PCO1) and 24% (PCO2) of the variability inherent in the resemblance matrix. However, in the PCO based on Presence/absence data (Data S5), less of the variability inherent in the resemblance matrix is reflected by the two main axes 28.1% and 16.8% respectively.

A posteriori pairwise PERMDISP test highlights that the RF zone conserve a homogeneous dispersion in variances before 1990s and 2019 data, whereas the CG zone shows heterogeneity in variance of data (*P* (tables): 0.59 and < 0.01 respectively; Data S4C). Therefore, the analyses indicate changes between period before 1990s and 2019 in the

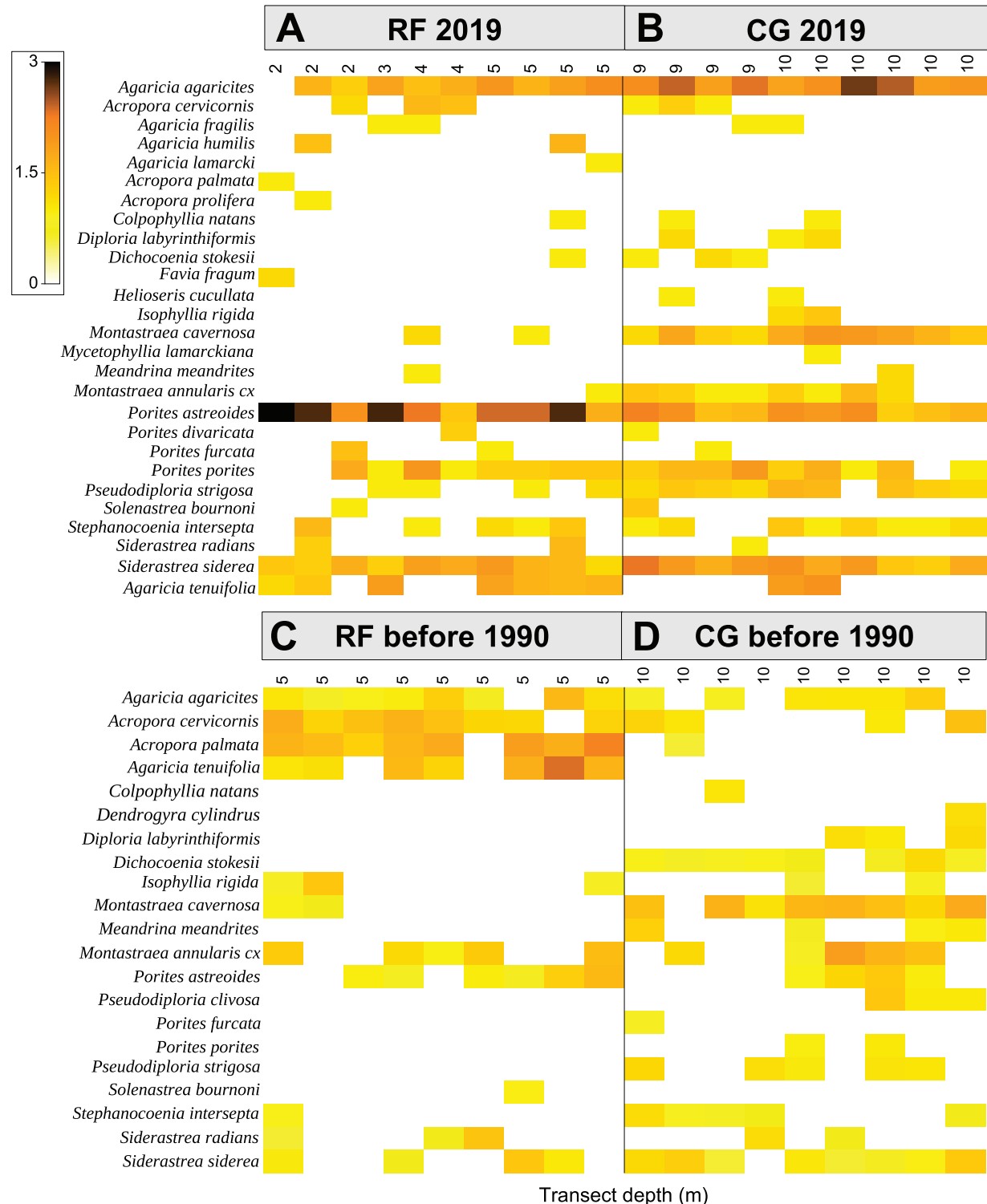

**Figure 4 Comparison of the contemporary and historical abundance for each species at transect level.** Color shaded plots of fourth root transformed species abundance (rows) by transect (columns) and zone (A and B: reef front RF and coral-ground, CG respectively for contemporary (2019) and (C and D): reef front, RF and coral-ground, CG for historical (before 1990s) data). (A) The (ribbon) colored scale is shown in the key with back-transformed counts where white squares indicate zero counts or species accounting for 5% or less of the total abundance. The *x* axis represents the transect depth (in meters) of n-samples. The *y* axis represents coral species.

**Table 2 Results of PERMANOVA and PERMDISP tests.** A synthesis of data transformation and pre-treatment procedure is presented for abundance and diversity data analyses. Bold font indicates statistically-significant differences (pairwise tests) or statistically significant interaction in the effects of factors. C1: Contrast analysis of factor year comparing 1979 vs. 1985; C2: Contrast analysis of factor year comparing pooled (1979/1985) vs. 2019; LIT: line intercept transect, BT: belt transect, CG: Coral-ground zone, RF: Reef front zone of the fringing reef; Some tests are highlighted with an (*) because the potential effect of data pooling in the test result (see section "Method" biases and uncertainties and Discussion for more information).

| Raw data type | 1979 and 1985: Relative abundance (live coral coverage %) 2019: Absolute abundance (number of individuals) | | | | | |
|---|---|---|---|---|---|---|
| Primary transform | 1979 and 1985: Standardize by total sample 2019: standardize by total sample | | | | | |

| Analysis | Abundance | | | Diversity | | |
|---|---|---|---|---|---|---|
| Overall transform | Square root of standardized data | | | Presence–absence | | |
| Resemblance | Bray–Curtis similarities matrix | | | Jaccard similarities matrix | | |

| Two-ways PERMANOVA | Pseudo-F | $p$ (perm) | $P$ (MC) | Pseudo-F | $p$ (perm) | $P$ (MC) |
|---|---|---|---|---|---|---|
| **Factor Year (3 levels: 1979, 1985, 2019)** | 13.77 | **<0.01\*** | **<0.01\*** | 5.45 | **<0.01** | **<0.01** |
| C1: Contrast 1979 vs 1985 (LIT) | 2.12 | 0.09 | 0.13 | 1.59 | 0.12 | 0.15 |
| C2: Contrast (1979/1985) vs 2019 (LIT vs BT) | 23.23 | **<0.01\*** | **<0.01\*** | 8.90 | **<0.01** | **<0.01** |
| **Factor Zone (2 levels: RF, CG)** | 18.27 | **<0.01** | **<0.01** | 11.15 | **<0.01** | **<0.01** |
| **Factor Year × Zone** | 3.16 | **<0.01** | **<0.01** | 1.86 | **<0.01** | **0.02** |
| C1: RF/CG 1979 vs RF/CG 1985 (LIT) | 1.18 | 0.46 | 0.41 | 0.78 | 0.66 | 0.59 |
| C2: RF/CG (79/85) vs RF/CG 2019 (LIT vs BT) | 4.49 | **<0.01\*** | **<0.01\*** | 2.77 | **<0.01** | **<0.01** |

| PERMANOVA (pair-wise tests) | $t$ | $p$ (perm) | | $t$ | $p$ (perm) | |
|---|---|---|---|---|---|---|
| RF 1979 vs CG 1979 (LIT) | 2.95 | **<0.01** | **<0.01** | 2.25 | 0.07 | **0.03** |
| RF 1985 vs CG 1985 (LIT) | 2.12 | **0.03** | **0.02** | 1.97 | **0.03** | **0.02** |
| RF 2019 vs CG 2019 (LIT) | 2.93 | **<0.01** | **<0.01** | 2.05 | **<0.01** | **0.01** |
| **PERMDISP (pair-wise tests )** | | | | | | |
| C2: CG (79/85) vs CG 2019 (LIT vs BT) | 6.22 | **<0.01\*** | | 4.59 | **<0.01** | |
| C2: RF (79/85) vs RF 2019 (LIT vs BT) | 0.08 | 0.95 | | 0.96 | 0.39 | |

composition of both zones (PERMANOVA) and in the variance of sample distribution (PERMDISP) in the CG zone.

## DISCUSSION

Regrettably, like in other Caribbean reefs, coral assemblages covering the reefal seascape at Punta Maroma have declined markedly in the past 35 years. Prior to the 1990s, there was a clear differentiation of coral assemblages between the shallow reef front (RF) and the deeper coral-ground (CG), with the former having higher presence of *Acropora* spp. and *Ag. tenuifolia*, and the latter with higher presence of small massive species such as *M. cavernosa, S. siderea, D. stokesii* and *Ag. agaricites*. These coral assemblages had become more homogeneous in abundance and species composition, with similarity among zones rising from 21.49%, before the 1990s, to 48.32%, by 2019 (Data S4B). Furthermore, by 2019, both the accretionary RF and the non accretionary CG community had the same coral species richness (S = 23), roughly the same colony density (2.4 per m$^2$) and were

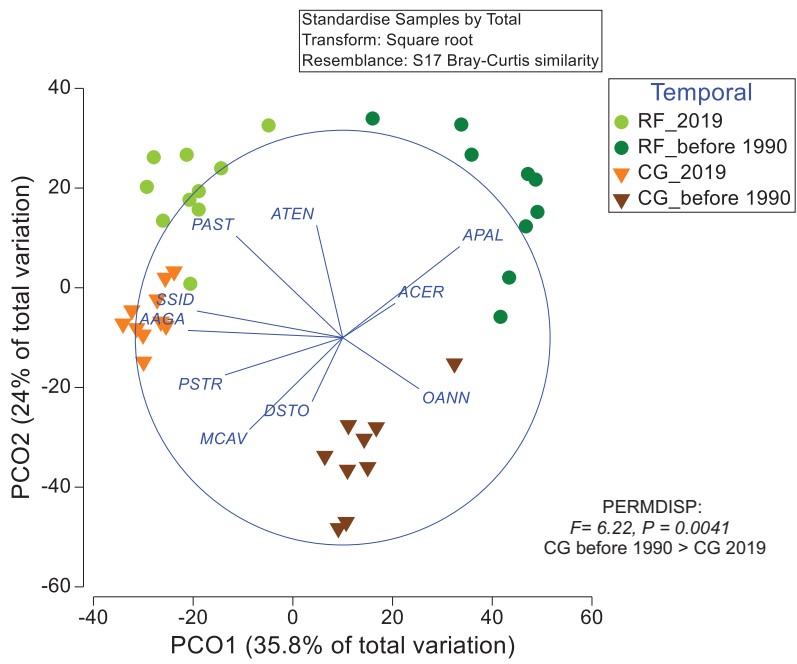

**Figure 5 Principal Component Analysis (PCO) derived from the Bray–Curtis similarity matrix constructed using a fourth root transformed matrix of standardized abundances of scleractinian coral species in two sampling zones at Punta Maroma reef seascape before 1990s and in 2019: a frontal zone of a fringing reef (RF) and a coral-ground (CG) zone.** Vectors visualize, through Pearson correlation coefficient, the potential monotonic relationship between the species accounting for 70% of total abundances and ordination axes a PCO. AAGA: *Ag. agaricites*, ATEN: *Ag. tenuifolia*, ACER: *Ac. cervicornis*, APAL: *Ac. palmata*, DSTO: *D. stokesii*, MCAV: *M. cavernosa*, OANN CX: *Orbicella* spp. complex, PAST: *Po. astreoides*, PSTR: *Ps. strigosa*, SSID: *S. siderea*.

dominated by small colonies (<20 cm) of *S. siderea, P. astreoides, Ag. agaricites*, and *P. porites*. However, according to the Index Value of Importance (IVI), reef builders like *Ac. palmata* conserved its high ecological value in the RF zone, despite its reduction in abundance through time, and the *Orbicella* (formerly *Montastrea*) reef-building group has a high value in the CG zone through time, despite the fact that this is a non-accretionary zone.

Reductions of live coral cover and decline in the abundance of large key reef-building species have been reported from other Caribbean reefs, together with the increase in the similarity of the coral assemblages and concomitant decrease in the visual difference amongst shallow adjacent coral-reef zones (*Gardner et al., 2003*; *Jackson et al., 2014*). Although the boundaries between these reef communities have often been defined by both biological and geological characteristics, these definitions have been inconsistent (e.g., ecozones in Fig. 1B). For example, *Estrada-Saldívar et al. (2019)* recently reported that similar ecological changes led to functional convergence and homogenization between back-reef and fore-reef sites along the north-east Yucatan, including the one at Punta Maroma. However they considered the RF and CG zones as a single "fore-reef" zone. A similar approach was followed by *Jordan et al. (1981)* to describe the windward zone of the entire Mexican Caribbean. Our findings show that on a more detailed scale,

however, this ecological homogenization is incomplete and that each geomorphic zone still retains differences. A good example are the PERMANOVA results of pairwise tests based on abundance data. Although these could be obscured by uncertainties related to the method bias and disparate nature of raw abundance metrics, differences are also supported by the sibling test based on presence-absence data, which is more reliable. Furthermore the homogenization of colony sizes by 2019 imply that the absolute-abundance data could be used as an estimate of coral coverage. Further indication that abundance values are involved are the lower levels of variability in the resemblance matrix illustrated by PCO axes in the presence/absence version compared to the sibling PCO axes in abundance. Thus, we show that changes in beta-biodiversity identified by biodiversity analysis are partially responsible for the main temporal differences. It may be that this "partial" homogenization results from a convergence in species succession within each geomorphic zone, as reported on other Caribbean reefs (*Curran et al., 1995*; *Aronson & Precht, 1997*).

As highlighted by the statistical analysis, ecological differences in species distribution and their relative importance between adjacent geomorphic zones may be related to long-term environmental processes. Despite changes in community structure, the RF zone still has an irregular substrate, with stumps of dead *A. palmata* and several acroporid spur-and-groove sets that slope up to the crest, and the CG zone is still a flat undulating rocky terrace crossed by shallow furrows and coral veneered ridges. These conditions favor the persistence of sediment-tolerant species, like *M. cavernosa, A. agaricites* (*Lasker, 1980*; *Erftemeijer et al., 2012*) and *S. siderea* in the CG zone, whereas in the RF zone the higher dominance of *P. astreoides*, which colonized space vacated by *A. palmata* may be a successional stage following disturbance. If this interpretation is correct then it highlights the importance of long-term adaptive responses of coral species to geomorphic substrates.

Although retaining some of their geomorphic character, these once easy to differentiate geomorphic zones are now more difficult to separate based on coral cover or other ecological indices. This difficulty stems from the functional loss of major reef builders such as Acroporids (e.g., *A. palmata, A. cervicornis*) which are largely responsible for long-term accretion in shallow Caribbean reefs (*Macintyre & Glynn, 1976*; *Blanchon et al., 2017*; *Toth et al., 2019*). These losses were likely driven by multiple strikes from major Hurricanes that crossed the study area (Allen in 1980; Gilbert in 1988; Emily in 2005 and Wilma in 2005), and their coincident timing with white band/pox epidemics that were decimating acroporids elsewhere (*Gladfelter, 1982*; *Lewis, 1984*; *Aronson & Precht, 2001*; *Bruckner, 2002*). Although there is debate over the proximate cause (hurricanes vs disease outbreaks) the result was the same: a partial convergence of shallow coral communities with a concomitant structural deterioration (*Jackson et al., 2014*; *Elliff & Silva, 2017*). At Punta Maroma the largest decline in Acropids had taken place by the mid-1980s, and no additional evidence of large-scale species succession has been reported since, although disturbances have not decreased (*Nyström, Folke & Moberg, 2000*; *Schutte, Selig & Bruno, 2010*; *Graham, Nash & Kool, 2011*; *Rioja-Nieto & Álvarez-Filip, 2019*). This rapid decline at Punta Maroma is likely related to a regional species succession stemming from the widespread mortality of *A. palmata* and *A. cervicornis*, and their

replacement by more successful representatives of the Agariciidae and Poritidae, as reported previously *Aronson & Precht (2001)*.

The rapid transition to a partially homogenized coral community at Punta Maroma today is inconsistent with the reef's Holocene record, implying the importance of these changes for the future accretion potential of the reef. But assessing the contribution of key reef-building species, such as *A. palmata*, in this accretion process based on their current ecological condition is a challenging exercise which depends on the type of ecological indicator used. Analysis of changes in species abundance and composition on a relatively short time-scale indicates a reduction in its contribution and an inferred loss in accretion potential. However, more complex measures than relative abundance, like the IVI analysis, indicates that some acroporids have retained their relative importance, highlighting the important contribution of this species to reef accretion. This is because the IVI includes other data such as colony size in addition to species abundances, and so gives a more complete picture. Nevertheless, such indices may still not provide an accurate picture of which species is important for accretion. For example, relative abundance data indicate that the CG zone now has more reef-building species, implying a higher accretion potential, despite the fact that CG zone has been repeatedly reported as non-accretional one (*Rodríguez-Martínez et al., 2011*; *Blanchon et al., 2017*). As a consequence, even the best-suited ecological indices of reef-accretion potential may not give accurate estimates unless the geomorphic context of coral communities is considered in more detail. Furthermore, although ecological studies may provide a detailed snapshot of historical timescales, they may not be fully representative of the long-term development in complex geomorphologically zoned reef structures (*Aronson & Precht, 1997*; *Bellwood et al., 2004*; *Bruckner, 2012*). For example, *Vallès, Oxenford & Henderson (2019)* recently demonstrate that each reef type has its own pattern of coral coverage and that between reef types there are differences not related to method bias. Similarly, intrinsic ecological patterns can also be identified with more detailed scrutiny of the geomorphic zonation because, as we have shown here, differences occur not just in coral coverage but in the taxonomic groups. Thus, by using geomorphic zones and other than coral coverage metrics, our results suggest the presence of within reef types (additional to those among reef types) ecological patterns.

Finally, the ecological dynamics of reef-building communities raises an interesting question about how they create geological structures over thousands of years. Ecological assessments assume that reef accretion potential is a function of the carbonate production rate of the site (gross production of primary and secondary producers less gross erosion of bioeroders (*Perry & Morgan, 2017*)), and therefore this balance defines the persistence of reef structure. Whereas those core processes undoubtedly lead to accretion, other processes are poorly represented, like physical erosion and transport of coral skeletons by storm swell or hurricane, and the patchiness and species zonation inherent to biological processes (*Perry, 1999*; *Purkis & Kohler, 2008*). Although it is challenging to incorporate physical erosion within the accretion potential estimates, geomorphic zonation is a more predictable factor with definable boundaries (*Blanchon, 2011*). Furthermore, the assumptions that reefs were always covered by dense coral thickets or

inhabited the same contemporary reef areas is questionable and ignores processes that exist outside of ecological timescales, for example, changes in sea level and the related processes of retrogradation or progradation of the reef structure (*Graus & Macintyre, 1989*). Indeed, little is known about the patterns of long-term accretion because geological reconstructions are largely two dimensional, deriving data from single drill holes or drill transects, and therefore assume accretion is a homogeneous process in space and time. It may be, for example, that accretion is heterogeneous in space and time and that some sections of a reef develop at different intervals, in different areas through time. In this case, some ecological conditions may not be representative of geological trends (*Jackson, 1992*).

## CONCLUSIONS

Over the last 40 years, coral assemblage data show that the two main windward geomorphic zones at Punta Maroma have maintained ecologic and benthic differences, implying that physical environmental drivers continue to exert a fundamental control on this reefal seascape. As a consequence, we suggest that a consideration of geomorphic zonation is a fundamental prerequisite for determining the accretion potential of entire reef systems. These results also indicate that there has been a partial homogenization of coral assemblages over that interval involving the loss of key reef-building species, which has raised concern about future accretion potential and the role of reef structures for coastal services. However, by considering the more detailed changes between geomorphic zones, our data do not rule out the possibility that accretion may in-fact be heterogeneous in space and time and that present-day coral communities may be the result of successional failure induced by chronic anthropogenic disturbance related to mass tourism along the Mayan Riviera.

## ACKNOWLEDGEMENTS

We are grateful with E. Perez-Cervantes and N. Estrada-Saldívar who participated in the coral surveys of 2019, and with L.M. Guzmán, A. González de la Parra and M. Sánchez who assisted in the survey of 1979 and 1985. I. Ortega, E.G. Islas Domínguez and F. Medellín assisted with data collection and logistics. Edlin Guerra Castro assisted greatly with multivariate statistical data analysis.

### Funding

Alexis E. Medina-Valmaseda was supported by a PhD scholarship (No. 666383) from CONACyT. Paul Blanchon was funded by the Mexican Council of Science and Technology (CONACyT, A1-S-18879) and the Universidad Nacional Autónoma de México (DGAPA/PAPIIT RN214819). The funders had no role in study design, data collection and analysis, decision to publish, or preparation of the manuscript.

## Grant Disclosures

The following grant information was disclosed by the authors:
CONACyT: 666383.
Mexican Council of Science and Technology (CONACyT): A1-S-18879.
Universidad Nacional Autónoma de México: DGAPA/PAPIIT RN214819.

## Competing Interests

The authors declare that they have no competing interests.

## Author Contributions

- Alexis Enrique Medina-Valmaseda conceived and designed the experiments, performed the experiments, analyzed the data, prepared figures and/or tables, authored or reviewed drafts of the paper, and approved the final draft.
- Rosa E. Rodríguez-Martínez analyzed the data, prepared figures and/or tables, authored or reviewed drafts of the paper, and approved the final draft.
- Lorenzo Alvarez-Filip performed the experiments, authored or reviewed drafts of the paper, and approved the final draft.
- Eric Jordan-Dahlgren analyzed the data, authored or reviewed drafts of the paper, and approved the final draft.
- Paul Blanchon conceived and designed the experiments, prepared figures and/or tables, authored or reviewed drafts of the paper, and approved the final draft.

## Field Study Permissions

The following information was supplied relating to field study approvals (i.e., approving body and any reference numbers):

This study did not involve the collection of samples or manipulation of the habitats, therefore a permit is not needed. However, we informed the authorities of Comision Nacional de Areas Naturales Protegidas (CONAMP) prior to conducting fieldwork as it is a requirement in order to carry out activities within MPA.

## Data Availability

The raw data and code are available in the Supplemental Files and at GitHub: https://github.com/AlexisMedina2019/Medina-Valmaseda_PeerJ_48068_Punta_Maroma_SuppData.

## Supplemental Information

Supplemental information for this article can be found online at http://dx.doi.org/10.7717/peerj.10103#supplemental-information.

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
