# Peer review of "The role of geomorphic zonation in long-term changes in coral-community structure on a Caribbean fringing reef"

_PeerJ, doi:10.7717/peerj.10103_

## Round 0.1 · original submission · Major Revisions

Dear Mr. Medina-Valmaseda,

Thank you for submitting your manuscript “The role of geomorphic zonation in long-term changes in coral-community structure on a Caribbean fringing reef” to PeerJ. I received reports from two reviewers, and as you will see below, both found your study to be interesting and well written and believe that it could be suitable for publication in PeerJ following revisions.

I have read the manuscript myself and concur with their assessments. Therefore, I ask that you revise your manuscript to address each of the comments raised. In particular, both reviewers had concerns about the temporal aspects of the data that I believe are quite relevant and should be addressed. Reviewer 1 also noted that they would like to see a more detailed explanation for the categorization of “Key species” vs. “Non-accretional species”. Although both the reviewer and I tend to agree with the classification used, given the manuscripts major emphasis on reef accretion I believe they are correct and a more detailed justification is warranted.

I look forward to seeing your revised manuscript.

Best,
Andy

·

Basic reporting

No comment

Experimental design

No comment

Validity of the findings

No comment

Additional comments

Review peerj-48068

General comments
The work by Medina-Valmaseda and collaborators entitled The role of geomorphic zonation in long-term changes in coral-community structure on a Caribbean fringing reef describes a homogenization process experienced by a Caribbean coral reef. In the last 40 years (1979-2019), two “geomorphic zones”, shallow (3.7 m) front reef, and deeper (9.8 m) rock-terrace reef has become more ecologically similar. The manuscript has good quality, it is well written, and the statistical methods are appropriated. The results are nicely presented but it needs some improvement before it can be accepted for publication. My main concerns are related to the overall geological approach, rather than ecological, evidenced by the use of geological terms (and the justification for it) as well as interpretation and discussion of the results. Below, I provide suggestions/comments for authors to consider in their review process.

The use of multiple terms related to ecological and geological reef processes is unclear and to some extent confusing. Take, for example, the geo zones and eco zones indicated in Figure 1. Throughout the manuscript multiple terms [(rock-terrace (line-97), coral ground (line 54), non-accretionary communities (line 67), non-accretionary deeper hard-coral ground (line 96)] are used referring to the same reef zone (hard-ground community, HG). Something very similar happens with shallow accretionary reef front, RF. Thus, while the combination of geological and ecological aspects to analyze geomorphic zones is novel and useful, it should be accompanied by a list of terms and their clear respective definitions (I suggest this definition list is submitted as supplementary material). More importantly, the use of these terms has to be standardized throughout the manuscript.

From the beginning and throughout the Introduction section, the authors focus on reef accretion, which they defined as a process occurring over geological time scale involving framework growth (e.g., CCA and corals) and erosion (physical and biological) as well as other factors (line 34-37). The introduction of the concept, accretion, provides the background for the distinction of the two geomorphic zones “accretionary zone” and “non-accretionary zone”. If “non-accretionary zone” is similar to “coral-ground”, already defined by Rodriguez-Martinez et al. (2011) as reef communities that colonize rocky substratum but not form framework (three-dimensional structures), why not use the coral-ground throughout the manuscript? It is a simpler term, already published with which readers might be more familiar with.

The authors indicate the use of ecological snapshots to estimate the accretion potential of the reef has two questionable assumptions (line 69-70) and one may wonder how this work, a temporal comparison of coral community snapshots data, deal with those two assumptions.

Did authors establish a size limit to differentiate juvenile (e.g., smaller than 4 cm) from adult corals? Since density (number of colonies m-2) and frequency were used in their analysis, readers need to know the minimum size considered during their surveys (at the very least those conducted in 2019)

Regarding the accretion approach, what are the biological bases to differentiate between Key species (usually coined Primary frame builders) and Less influential species (traditionally called secondary frame builders)? If accretion involves physical and biological processes occurring over a long-time period that determines the balance between reef growth and erosion how then, can some species (slow-growing, branching, and massive) be classified as Key species and others (fast-growing, less structurally complex) as less influential? One could imagine that besides species growth rate, colony size, and structure, other elements might be taken into account to distinguish their role in reef accretion. A better explanation for the accretion Key species vs. accretion less influential species will greatly enhance the paper.

One of the most straightforward results from this temporal analysis is the homogenization process experience by the coral communities present in the two geomorphic zones. Yet, the discussion and conclusion mostly focus on the potential accretion (line 308, 330). I suggest the authors bring to the discussion, and maybe introduction, how they differentiate between ecological processes such as ecological succession and ecosystem resilience vs. reef accretion. For example, in line 42-44 …”Left undisturbed, this framework has the potential to accrete vertically”……a clarification of what kind of disturbances will help to understand the outcome “accrete vertically”.

Specific comments
Line 16: This first sentence of the abstract is confusing. Are the authors referring to Ecological processes or Ecological studies (approaches)?

Line 20: Define geomorphic structure

Line 39: Cite a couple of examples where CCAs are among the primary reef calcifier [examples, algal ridges of St. Croix. (Adey 1975), Atoll Rocas in Brasil (Kikuchi et al. 1997), Hawaiian fringing reefs (Littler 1973)].

Line 48-49: It’ll be more accurate to say “For example, analysis of cores drilled on Caribbean …..”

Line 51-52: Same comment as above: “…, in more hurricane-prone areas, cores obtained in the same reef zones….”

Line 53-55: This sentence is confusing and should be modified. Assuming “non-accretionary coralgal framework” is the synonym of “coral ground”, I suggest the author use the term “coral ground” since it’s been defined in the past (e.g., Rodriguez-Martinez et al. 2011) and readers might be more familiar with it.

Line 56: Eliminate “natural”

Line 67-68. In line 67 the “non-accretionary” term is defined as “communities” whereas in line 68 is considered is zone “geomorphic zones”. As stated before, I suggest the authors choose a term (framework, community, or zone) for non-accretionary areas and stick with it. A solution could be to “coral-ground zone”.

Line 145: Replace “This is done because different geomorphic zones within a reef have a heterogeneous accretion capacity due not only to the composition of the coral community but also to external environmental gradients” with “The analysis of IVI was carried out because different geomorphic zones within a reef have a heterogeneous accretion capacity as a consequence of the composition of the coral community and external environmental gradients”

Line 182: Replace “genres” with “genera”

Line 186: Replace “genres” with “genera”

Line 190: Eliminate “the cornerstone”

Line 207. Since A. palmata, A. cervicornis, A. tenuifolia, and A. agaricites refer to two distinct genera, the genus acronym must be differentiated using, in this case, the first and second letter. Ac palmata, Ac. Cervicornis, Ag. Tenuifolia, and Ag. Agaricites.

Line 275-277: Please, add references supporting that species of Agaricia and M. cavernosa are sediment-tolerant species.

Line 306-307: Please, add the reference support this claim “….., despite the fact that geological data indicate no accretion during the Holocene”

Line 314-315: The statement “Ecological assessments assume that reef accretion is constant in time and space” can be argued over the basis what the goals of geological vs. ecological studies. I suggest the authors elaborate on it by bringing up examples illustrating their points.

Line 330: Add “,” after “consequence”

Tables and figures
Table 1. There is no need for this table since most of the information has been already included in the text [“No of transects” (line 103), ‘S” (line 182, 185), and density (line 181)]. “Average depth” can be mentioned in the Methods section and “N”, if needed, can be mention in the Results section.

Figure 3 and figure 4. I suggest, if possible at all, to use short written species name (S. siderea) instead of species acronym (SSID).

References
Adey, W. H. (1975). The algal ridges and coral reefs of St. Croix, their structure and Holocene development. Atoll Res. Bull. 187, 1–66. doi: 10.5479/si.00775630.187.1

Kikuchi, R. K. P., and Leão, Z. M. A. N. (1997). “Rocas (southwestern equatorial Atlantic, Brazil): an atol built primarily by coralline algae”in Proceedings Of The International Coral Reef Symposium, Panama, 731–736.

Littler, M. M. (1973). The population and community structure of Hawaiian fringing-reef crustose corallinaceae (Rhodophyta, Cryptonemiales). J. Exp. Mar. Biol. Ecol. 11, 103–120. doi: 10.1016/0022-0981(73)90050-6

Rodriguez-Martinez, R. E., Jordan-Garza A. G., Maldonado M. A., Blanchon P. (2011). Controls on coral-ground development along the Northern Mesoamerican Reef Tract. PLoS ONE 6(12): e28461

Reviewer 2 ·

Basic reporting

The manuscript is well written and fulfills the necessary conditions of basic reporting.

Experimental design

It would be good to provide some more detail on how the 2019 transect data were handled - for example, it is not clear whether their abundance estimates are based upon colony counts or colony surface area (unless I have missed it). Otherwise, I find it to be of good standard.

Validity of the findings

This is where I have my main concern - The authors combine data collected using fairly different benthic protocols but provide no clear justification as to why this is appropriate. It is thus not clear to which extent their results might be confounded by method biases. Please see my comments for the authors for more detail.

Additional comments

Overall, I found this to be a well-written manuscript in an important topic. There is great need to rigorously assess the extent to which coral reefs have changed over time. The authors focus on a fringing reef at Punta Maroma to assess changes in coral assemblages over a 40-year period. To do so, the make use of data collected in the 1979 and 1985 as baseline and compare these data with those collected in 2019 at similar reef zones on the same reef. They use a range of multivariate statistical tools to carry out such comparisons and finally conclude that there is strong evidence of a partial homogenization of what used to be distinct Reef front and Hard ground coral communities.
The manuscript is very well written, and the authors clearly show good mastery of the statistical techniques they use. However, there is a fundamental problem underlying all their analyses, which compromises to an unknown extent the robustness of their conclusions. The 1979/1985 data used as baseline were collected using a very different sampling protocol from the data collected in 2019. In 1979/1985 data were collected using 20-m long line transects and counting the chain links overlaid of different coral species. The 2019 dataset involved using 30-m long (1-m wide) belt transects and counting and measuring the corals within those transects. Yet, in their statistical analyses the authors treat these very different sampling units as perfectly interchangeable without a clear justification of why this should be the case. They allude to a contrast-based analysis and to the fact that both data sets yielded increasingly flattening accumulation curves, but it is not clear how these points address the issue of method bias and interchangeability of statistical replicates. In my view, flattening accumulation curves do not imply absence of method-bias and it is not explicitly clear how their contrast-based analysis addresses the issue. Yet, it seems unreasonable to simply assume that sampling units of such different spatial grain sizes (line transect: 20 m x 0.01 m = 0.2m2 vs belt transect: 30 m x 1m = 30m2) applied on the same reef zone would necessarily exhibit similar statistical properties (e.g. variance and bias). As Fieberg et al (2020) recently point out (https://peerj.com/articles/9089/), the resampling methods that underlie the multivariate analyses performed by the authors assume that the sampling distribution of a statistic (e.g. Pseudo-F) “tells us about the values we might expect for the statistic if we were to repeatedly collect data sets of the same size from the same population and using the same sampling protocols”. The latter condition is clearly not met in the authors’ study. For all we know, some of the multivariate dispersion differences identified by the authors between periods (lines 244-245) could simply reflect reef zone-specific differences between methods in sample variance.
I have recently become increasingly aware of the issue of method-bias in coral reef surveys and my view is that this remains a pervasive problem in the literature, despite a considerable number of studies highlighting that it should not be ignored (for examples addressing the issue of method-bias, see https://peerj.com/articles/8167/ and references therein). Yet, it is currently not being sufficiently recognized and adequately addressed in studies seeking to compare or combine datasets collected using different methods. I also feel it is solely the responsibility of the authors to show that the probability of method-bias confounding their analyses is negligible and/or can be explicitly dealt with. Ideally, the authors should have carried out a pilot study using both methods in 2019 to demonstrate negligible or manageable method-bias to help justify their data pooling. Unfortunately, this preliminary step is hardly ever considered in the literature. Alternatively, the authors could at least point to empirical and/or theoretical studies focusing on comparisons between line transects and belt transects supporting that it is adequate to pool the data. Note that this would be obviously different from pointing to prior published studies suffering from the same problem.
Having said that, I do not question the main qualitative conclusion of the authors - I am sure that, overall, the reef zones have changed in the general direction of homogenization highlighted by the authors - My concerns revolve more around the technical soundness of the analyses allowing them to reach those conclusions and around the potential for method-biases to consistently over- or under-estimate the true magnitude of change.
Thus, I think the authors should be given an opportunity to provide an adequate justification for their data pooling, which should go well beyond what is currently offered in the manuscript, and which should be made explicit in the manuscript itself. However, if the latter is not possible, my recommendation to the authors would be to simply refrain from any statistical comparison between the 1979/1985 period and the 2019 that would assume that their data are perfectly interchangeable between those periods. The manuscript could use both datasets independently to illustrate the coral communities during each period. Most of their outputs already seem to do so (Table 2, Fig 2-4) and they could perhaps be supplemented with species richness estimators and accumulation curves for each dataset. In any case, the changes reported appear so strong that I am sure that the final take-home message will remain clear and convincing without having to compromise the overall statistical soundness of the paper.

---

## Round 0.2 · Minor Revisions

Thank you for previous revisions. As you will see, both the reviewer and I believe the manuscript is improved. As you will see, Reviewer 2 has noted a number of minor corrections that should be made after which I believe your manuscript will be suitable for publication.

Best,
Andy

Reviewer 2 ·

Basic reporting

The paper is generally well-written, but there are a few instances where sentences would benefit from some revision. I have pointed out these in the specific comments.
I felt the Intro would benefit by ending with the formulation of a clear and explicit objective.
Figures and tables are generally fine, minus some text omissions/typos. I have pointed out these in the specific comments.

Experimental design

I have one significant additional concern in relation to the handling of the data used to compare changes in "relative abundance" between 1979/1985 and 2019 - I provide details in the comments below. Having said that, I do not think that this concern will ultimately affect the authors' main conclusions; this concern speaks more to the technical soundness of the paper.

Validity of the findings

No further comment

Additional comments

General comments
Overall, I feel this version is much improved and that it has addressed the main issue about method-biases I had raised in my previous comments. There is also greater clarity in how the data were handled and analyzed. Because of the latter, there is one potentially additional significant issue that has caught my attention, which is the use of number of colony counts in the 2019 dataset and the number of chain links in the 1979/1985 to derive estimates of relative abundance to compare between periods. My sense is that numbers of chain links inform mainly about spatial living coverage of the different coral species, irrespective of their colony numbers, so it is not quite clear that these two metrics are directly comparable. In my view, a more meaningful comparison would have involved deriving estimates of relative abundance based on actual colony counts for both periods (since these data seem available) OR based on surface area covered by each coral in 2019 (since the data on both colony counts and colony size also seem available) and number of chain links in 1979/1985 period. Having said that, the authors have supplemented their key analyses with presence/absence data, which should be fairly robust to these data handling decisions, and the general patterns seem to hold (Table 2). As such, I would endorse the manuscript for publication after minor revisions provided that the authors give appropriate consideration to my specific comments below.

Specific comments
Line 50-53 – It is not clear what does the “however” make reference to – I get the sense that the point here is that you can have coral grounds that are not geologically accretionary because they have been subject to high levels of physical erosion by hurricanes and they are thus basically made of coral fragment remains. If that is what is meant, introducing the term non-accretionary in that sentence would help as a contrast to the previous sentence.
Line 80-83 – This sentence is not clear – are the surveys the ones in the paper or those from a different study? Moreover, the objective is not very clear – I presume that it is to assess the extent to which homogenization has occurred in these two geomorphologically different zones? It would be good to mention that historical data from the 70’s will be compared to recent data for that purpose to inform about the specific temporal period of interest.
Line 133 –comma should come after citation brackets
Line 134 – after b), sentence should star in lower case
Line 142 – Please clarify that the IVI for the pre-1990’s data is only based on the 1985 dataset.
Line 153-154 – I think the key idea here is that if the method-biases are small relative to the real signal of change, then the latter would still be reliably and meaningfully detected. The Nadon and Jokiel references could then be used to argue that previous studies comparing the two methods indicate that such method biases appear indeed to be small - the later conceptual framework (i.e. method biases being small relative to the true signal) is probably better because using the term “virtually indistinguishable” might require clarification about whether or not their tests had sufficient power, which is often not the case.
Line 163-167 - Good. I also agree that diversity metrics are likely to be less affected by method-biases than % coverage ones.
Line 172 – replace “…which has difficulties dealing with this using …” with “…which makes it difficult to deal with using …”
Line 175 – 185 - I feel that the method-bias issue has been sufficiently addressed in the “Method biases and uncertainties section” and I do not think that any further action is needed. In the previous section the authors provide sufficient arguments to conclude that, although method-biases are likely, the main choice of metrics by the authors (i.e. diversity metrics), along with findings from other studies cited comparing the two specific methods at hand, suggest that these method-biases are likely to be small relative to the effects that the authors are investigating in this particular study. It is not perfect, but it is probably enough to help justify their statistical approach and it is much better than not saying anything about potential method-biases (as it was in the previous MS version).
The problem I see now in this section (lines 175-185) is that it provides a sense that method-biases can be somehow dealt with once the data have been collected. This is not true unless it is via the use of calibration/conversion curves linking both methods, which the authors do not have. I would thus recommend removing lines 175-179 and any subsequent reference to method-biases in this section. The various standardizations of the data can probably remain as they are.
Line 180-182 – Why are colony counts used to estimate relative abundance in the 2019 dataset considering that chain links are used to do the same in the 1979/1985 dataset? Using colony counts does not seem to consider that different coral species differ markedly in size. You can have a small-sized coral species (A. agaricites, P. astreoides) scoring very high in relative abundance even though overall it covers much less living space than a few but bigger coral species (Orbicella spp). This clarification is important because using the chain links to estimate relative abundance in the 1979/1985 datasets will better reflect the spatial living coverage of the different species than the actual number of colonies (unless I am missing something) [see Loya 1972 Mar Biol 13:100-122 as an example of a study using line transects to calculate both relative abundance (based on no of colony counts under a chain) and living coverage (based on no of chain links under the chain) of a coral community]. It would have made more sense to transform the 2019 dataset in relative spatial living coverage data (rather than relative abundance data), since the colonies were counted and sized, prior to calculating their relative abundance to compare with the 1979/1985 dataset. Alternatively, the colony counts of the 1979/1985 data could be used (instead of the chain links) to calculate the relative abundance of each species to compare with the 2019 dataset (as in Loya 1972). As it stands now, it might seem like comparing apples (2019: colony count estimates) with oranges (1979/1985: living coverage index), which if true could contribute to artificially create differences between time periods. I suspect I might not be the only one wondering about this and so this data handling decision will require further clarification. Perhaps one draconian (but more robust) way to deal with this potential problem would be to focus on presence/absence data only throughout the manuscript - this approach is already done in the “Diversity” column of Table 2; presence/absence data could also be used for Figures 4 and 5.
Line 220 – consider removing the commas
Line 243- consider removing the commas
Line 255 – Replace “Montastrea annularis complex” with “Orbicella…”
Fig 2 – clarify in figure header that data from 1979 and 1985 were pooled together in panels C and D.
Line 270 – insert “the” before “two zones”
Line 273 -274 – Please clarify - is it correct to use the value of the Pseudo-F ratio as a measure of effect strength when comparing among PERMANOVA factors? Does Pseudo-F ratio value not depend on the degrees of freedom (and/or levels) associated with each factor, which is likely to differ across factors?
Figure 4 – Is it possible to show the same species rows across all four panels to facilitate cross-panel comparisons in all directions? Please see my previous comments (on lines 180-182) about what these abundance estimates actually likely reflect for each method/time period. Also, it should be “…fourth root…” in header text.
Table 2 – Please clarify that normal (not in bold) font in under the two-way PERMANOVA header corresponds to the contrasts – I presume that the lines under the Factor Year x Zone also correspond to contrasts, but these (unlike the ones above under Factor Year) were never mentioned in the Methods (but I might have missed it) and it is not quite clear what they represent – moreover, one line has “RF,CG 1979” and the line below “RF/CG (79/85)” –not clear how the use of “,” versus “/” is to be interpreted. Also, something seems wrong with the last two lines under the PERMANOVA (pair-wise tests) header (repetition of terms). Finally, the 2nd and 3rd sentence of the Table 2 header would benefit from some re-phrasing.
In relation to the PermDisp tests, is it necessary to do all pair-wise comparisons? Why not simply focus on the within-zone temporal comparisons (the 3rd and 4th lines under the header)? It would facilitate digesting these results. Also, how does the fact that the variance in CG differs between pre-1990’s and 2019 affect interpretation of the PERMANOVA results for CG, which presumably still assume similar dispersion between time periods when comparing centroids?
Fig 5 – Note that the permdisp value in the figure does not correspond with the one in the table. Replace “Montastrea annularis complex” with “Orbicella annularis…”
Line 294-306 – I think it is important to acknowledge that (1) both zones have changed markedly over time, and (2) that they done so in a manner that has led to some homogenization. I think this is the best interpretation of Fig 5 (i.e. it not only about homogenization, which could imply only one zone changing).
Line 300 – I do not recall any reference to these values (21.9% and 52.5%) in the Results section or supplementary material, but I might have missed it.
Line 313 – it might be important to distinguish between functional, taxonomic and ecological homogenization even from the Introduction and clarify which of these the MS is dealing with.
Line 318-319 – This idea is good and further developed in Line 326-335 – it might be better to integrate these two into a single paragraph at once – the Lines 320-250 dealing with the robustness of the results, which is an important consideration, could then come at the end or in a different paragraph.
Line 320 – I agree that any analysis based on the presence-absence data will be more reliable. As a suggestion, consider doing a similar figure to Fig 5, but based on presence/absence data to include as supplementary material.
Line 344 – consider adding the word “partial” in front of convergence.
Line 349 – this line would benefit from further development or details on the “regional species succession reported by Aronson and Precht (2001)”; enough so that the reader does not need to go to the cited paper to get a clearer sense of what it is implied.
Line 371-372 - Please clarify - Not clear if the contrast with Valles et al refers to identifying ecological patterns within reef types (rather than among reef types) or by using geomorphic zones or metrics other than coral coverage or a combination of any of these.
Line 376 – missing bracket.
Line 397-399- I was not sure about the basis for this statement about recruitment failure –it would benefit from further development or clarification in the Discussion or in the Conclusion itself.

---

## Round 0.3 · accepted · Accept

After reviewing the newest version of the manuscript I believe that you and your coauthors have adequately addressed all of the reviewers concerns. Congratulations, I appreciate the hard work that you put into this manuscript and believe it will make a very nice contribution to PeerJ.

Best,
Andy

---

## Author Rebuttal · Round 0.3

POSGRADO EN CIENCIAS DEL MAR Y LIMNOLOGÍA
INSTITUTO DE CIENCIAS DEL MAR Y LIMNOLOGÍA.UNAM

Dear Editor

We thank you and the anonymous reviewer for the comments on the manuscript. Again all the points raised were valuable and gave us the opportunity to improve the manuscript. The main concern of the reviewer was about the temporal analysis of pooled abundance data within the two-way PERMANOVA design. We understand the reviewer's concern and have edited the manuscript to provide a detailed justification about our statistical approach within the text and to the reviewer.

The main changes to the manuscript include adding a new figure to supplementary material under the recommendation of the reviewer, making minor corrections in tables and in the redaction of the manuscript, and changes in the order of coral species in figure 4 to facilitate cross-panel comparisons in all directions. The changes made to the previously submitted manuscript are described in detail below.

We hope that this updated version of the manuscript is now suitable for publication in PeerJ.

MSc. Alexis E. Medina Valmaseda
Postgraduate Program - PCMyL UNAM
On behalf of all authors

**General comments**

Overall, I feel this version is much improved and that it has addressed the main issue about method-biases I had raised in my previous comments. There is also greater clarity in how the data were handled and analyzed. Because of the latter, there is one potentially additional significant issue that has caught my attention, which is the use of number of colony counts in the 2019 dataset and the number of chain links in the 1979/1985 to derive estimates of relative abundance to compare between periods. My sense is that numbers of chain links inform mainly about spatial living coverage of the different coral species, irrespective of their colony numbers, so it is not quite clear that these two metrics are directly comparable. In my view, a more meaningful comparison would have involved deriving estimates of relative abundance based on actual colony counts for both periods (since these data seem available) OR based on surface area covered by each coral in 2019 (since the data on both colony counts and colony size also seem available) and number of chain links in 1979/1985 period. Having said that, the authors have supplemented their key analyses with presence/absence data, which should be fairly robust to these data handling decisions, and the general patterns seem to hold (Table 2). As such, I would endorse the manuscript for publication after minor revisions provided that the authors give appropriate consideration to my specific comments below.

*Reply: We thank the reviewer for the detailed and thorough review of our paper. Regarding temporal analysis of the abundance data, which we believe is the main source of the concern, we have expanded our considerations in the response to the comment 2.0 below and in the discussion (line 326-332).*

**Specific comments**

**Comment 1.1:** Line 50-53 – It is not clear what does the "however" make reference to – I get the sense that the point here is that you can have coral grounds that are not geologically accretionary because they have been subject to high levels of physical erosion by hurricanes and they are thus basically made of coral fragment remains. If that is what is meant, introducing the term non-accretionary in that sentence would help as a contrast to the previous sentence.

*Reply 1.1: We agree and have added the term.*

**Comment 1.2:** Line 80-83 – This sentence is not clear – are the surveys the ones in the paper or those from a different study? Moreover, the objective is not very clear – I presume that it is to assess the extent to which homogenization has occurred in these two geomorphologically different zones? It would be good to mention that historical data from the 70's will be compared to recent data for that purpose to inform about the specific temporal period of interest.

*Reply 1.2: We agree and have modified the text (lines 80-83)*

Comment 1.3:Line 133 –comma should come after citation brackets

*Reply 1.3: Done*

**Comment 1.4:**Line 134 – after b), sentence should star in lower case

*Reply 1.4: Done*

**Comment 1.5:**Line 142 – Please clarify that the IVI for the pre-1990's data is only based on the 1985 dataset.

*Reply 1.5: We agree and have modified the text (lines 142-143)*

**Comment 1.6:**Line 153-154 – I think the key idea here is that if the method-biases are small relative to the real signal of change, then the latter would still be reliably and meaningfully detected. The Nadon and Jokiel references could then be used to argue that previous studies comparing the two methods indicate that such method biases appear indeed to be small - the later conceptual framework (i.e. method biases being small relative to the true signal) is probably better because using the term "virtually indistinguishable" might require clarification about whether or not their tests had sufficient power, which is often not the case.

*Reply 1.6: We agree and have modified the text accordingly ( lines 156- 161)*

**Comment 1.7:**Line 163-167 - Good. I also agree that diversity metrics are likely to be less affected by method-biases than % coverage ones.

**Comment 1.8:**Line 172 – replace "…which has difficulties dealing with this using …" with "…which makes it difficult to deal with using …"

*Reply 1.8: Done*

**Comment 1.9:**Line 175 – 185 - I feel that the method-bias issue has been sufficiently addressed in the "Method biases and uncertainties section" and I do not think that any further action is needed. In the previous section the authors provide sufficient arguments to conclude that, although method-biases are likely, the main choice of metrics by the authors (i.e. diversity metrics), along with findings from other studies cited comparing the two specific methods at hand, suggest that these method-biases are likely to be small relative to the effects that the authors are investigating in this particular study. It is not perfect, but it is probably enough to help justify their statistical approach and it is much better than not saying anything about potential method-biases (as it was in the previous MS version).The problem I see now in this section (lines 175-185) is that it provides a sense that method-biases can be somehow dealt with once the data have been collected. This is not true unless it is via the use of calibration/conversion curves linking both methods, which the authors do not have. I would thus recommend removing lines 175-179 and any subsequent reference to method-biases in this section. The various standardizations of the data can probably remain as they are.

*Reply 1.9: We agree and have modified the text accordingly and removed the reference to method-biases in this section ( lines 179-180).*

**Comment 2.0:** Line 180-182 – Why are colony counts used to estimate relative abundance in the 2019 dataset considering that chain links are used to do the same in the 1979/1985 dataset? Using colony counts does not seem to consider that different coral species differ markedly in size. You can have a small-sized coral species (A. agaricites, P. astreoides) scoring very high in relative abundance even though overall it covers much less living space than a few but bigger coral species (Orbicella spp). This clarification is important because using the chain links to estimate relative abundance in the 1979/1985 datasets will better reflect the spatial living coverage of the different species than the actual number of colonies (unless I am missing something) [see Loya 1972 Mar Biol 13:100-122 as an example of a study using line transects to calculate both relative abundance (based on no of colony counts under a chain) and living coverage (based on no of chain links under the chain) of a coral community]. It would have made more sense to transform the 2019 dataset in relative spatial living coverage data (rather than relative abundance data), since the colonies were counted and sized, prior to calculating their relative abundance to compare with the 1979/1985 dataset. Alternatively, the colony counts of the 1979/1985 data could be used (instead of the chain links) to calculate the relative abundance of each species to compare with the 2019 dataset (as in Loya 1972). As it stands now, it might seem like comparing apples (2019: colony count estimates) with oranges (1979/1985: living coverage index), which if true could contribute to artificially create differences between time periods. I suspect I might not be the only one wondering about this and so this data handling decision will require further clarification. Perhaps one draconian (but more robust) way to deal with this potential problem would be to focus on presence/absence data only throughout the manuscript - this approach is already done in the "Diversity" column of Table 2; presence/absence data could also be used for Figures 4 and 5.

*Reply 2.0: We accepted the reviewer's suggestion and have performed the temporal analysis of abundance data after transforming absolute abundance from 2019 data into relative coverage to reduce the disparate nature of raw abundance data in time (line 182-185). These approaches did not alter the results of temporal comparative analysis based on contrasts for factor year. The reviewer also suggests a focus on presence/absence data throughout the MS, however, the concern about analysis of pooled abundance data only affects the temporal comparison, whereas abundance-based pairwise analyses of factor zones are uncompromised.*

*Moreover, from the perspective in which the MS was conceived, the zonal analyses are essential. As a consequence, we consider that both abundance and presence-absence results are complementary, not mutually exclusive. We reiterate that the main goal of the MS is the comparative analysis between geomorphic zones (see lines 76-78, 166-169 and 204-206) and we therefore compare the historical and contemporary data separately. In those pairwise analyses, we neither combine nor mix benthic methods. For the historical data (pre 1990s: 1979 and 1985) we only used the chain method, while for the 2019 year we only used the data collected from belt transects. In table 2 (PERMANOVA pairwise tests) we present the results in this manner and also highlight the benthic method involved in each case). We tackled the remaining issue of method bias and disparate nature of raw data extensively in the 'Method*

*Biases and Uncertainties' section, and expanded our justification for dual abundance/diversity approach in the discussion (324-332)*

**Comment 2.1** Line 220 – consider removing the commas

*Reply 2.1: Done*

**Comment 2.2** Line 243- consider removing the commas

*Reply 2.2: Done*

**Comment 2.3** Line 255 – Replace "Montastrea annularis complex" with "Orbicella…"

*Reply 2.3: Done*

**Comment 2.4** Fig 2 – clarify in figure header that data from 1979 and 1985 were pooled together in panels C and D.

*Reply 2.4: Done*

**Comment 2.5** Line 270 – insert "the" before "two zones"

*Reply 2.5: Done*

**Comment 2.6** Line 273 -274 – Please clarify - is it correct to use the value of the Pseudo-F ratio as a measure of effect strength when comparing among PERMANOVA factors?, which is likely to differ across factors?

*Reply 2.6: We have now corrected the manuscript and used the correct estimator for the effect strength under our statistical design (random/fixed multi factors: Estimates of components of variations; Underwood & Petraitis 1993 and Anderson 2017).*

**Comment 2.7** Figure 4 – Is it possible to show the same species rows across all four panels to facilitate cross-panel comparisons in all directions? Please see my previous comments (on lines 180-182) about what these abundance estimates actually likely reflect for each method/time period. Also, it should be "…fourth root…" in header text.

*Reply 2.7: Done*

**Comment 2.8** Table 2 – Please clarify that normal (not in bold) font in under the two-way PERMANOVA header corresponds to the contrasts – I presume that the lines under the Factor Year x Zone also correspond to contrasts, but these (unlike the ones above under Factor Year) were never mentioned in the Methods (but I might have missed it) and it is not quite clear what they represent – moreover, one line has "RF,CG 1979" and the line below "RF/CG (79/85)" –not clear how the use of "," versus "/" is to be interpreted. Also, something seems wrong with the last two lines under the PERMANOVA (pair-wise tests) header (repetition of terms). Finally, the 2nd and 3rd sentence of the Table 2 header would benefit from some re-phrasing.

***Reply 2.8:*** *We agree and have modified table 2 and its header in the manuscript to include the recommendations.*

Comment 2.9 In relation to the PermDisp tests, is it necessary to do all pair-wise comparisons? Why not simply focus on the within-zone temporal comparisons (the 3rd and 4th lines under the header)? It would facilitate digesting these results. Also, how does the fact that the variance in CG differs between pre-1990's and 2019 affect interpretation of the PERMANOVA results for CG, which presumably still assume similar dispersion between time periods when comparing centroids?

***Reply 2.9:*** *We have modified table 2 to address this comment. Regarding reviewer's concern about how PERMDISP results affect interpretation of PERMANOVA we would like to highlight that PERMDISP do not invalidate PERMANOVA, rather PERMDISP is a complementary test that aid the interpretation of the PERMANOVA results. It means that in addition to the differences in centroids there are also differences in dispersion (verify in the mMDS below that the difference is clear)(verify in the mMDS below that the difference is clear) (verify in the mMDS below that the difference is clear) . Both analyzes indicate that there was a change in the composition (PERMANOVA result) and heterogeneity in distribution (PERMDISP result) of the coral species in those geomorphic zones from 1979 (85) to 2019 (lines 298-300).*

[Figure]

Bootstrap Averages Plot of Bray-Curtis Similarity matrix of square root transformed abundance data previously standardized by Total

Number of bootstraps per group: 100. Minimum rho: 0.99. Bootstrap regions: 95%. Optimum m: 8  rho: 0.993  / Parameters: Kruskal stress formula: 1

Minimum stress: 0.01 / Stress values: 3D: 0.05, 2D: 0.09

**Comment 3.0** Fig 5 – Note that the permdisp value in the figure does not correspond with the one in the table. Replace "Montastrea annularis complex" with "Orbicella annularis…"

***Reply 3.0:*** *Agreed. This is now amended in the MS*

**Comment 3.1** Line 294-306 – I think it is important to acknowledge that (1) both zones have changed markedly over time, and (2) that they done so in a manner that has led to some homogenization. I think this is the best interpretation of Fig 5 (i.e. it not only about homogenization, which could imply only one zone changing).

*Reply 3.1: Agreed. That is the sense of those lines and we further specify in which way the zones have changed.*

**Comment 3.2** Line 300 – I do not recall any reference to these values (21.9% and 52.5%) in the Results section or supplementary material, but I might have missed it.

*Reply 3.2: Agreed. This has been amended now in the MS (line 303) and referring those values in the supplementary material (Data S4. B)*

**Comment 3.3** Line 313 – it might be important to distinguish between functional, taxonomic and ecological homogenization even from the Introduction and clarify which of these the MS is dealing with.

*Reply 3.3: Agreed. Throughout the manuscript we repeatedly allude to the constraints of detected partial homogenization (only in colony size and some ecological indexes) but not in taxonomic groups which we consider have implications for the geomorphological and geological approach of ecological studies. We consider this is solved in the discussion.*

**Comment 3.4** Line 318-319 – This idea is good and further developed in Line 326-335 – it might be better to integrate these two into a single paragraph at once – the Lines 320-250 dealing with the robustness of the results, which is an important consideration, could then come at the end or in a different paragraph.

*Reply 3.4: We accepted reviewer suggestion and modified the text in the manuscript to include this recommendation (lines 327-338)*

**Comment 3.5** Line 320 – I agree that any analysis based on the presence-absence data will be more reliable. As a suggestion, consider doing a similar figure to Fig 5, but based on presence/absence data to include as supplementary material.

*Reply 3.5: We accepted the recommendation, and have added the Figure as supplementary material Data S5 PCO Presence absence. This figure is now referred to in the MS in line 283 and 292- 294.*

**Comment 3.6** Line 344 – consider adding the word "partial" in front of convergence.

*Reply 3.6: Done*

**Comment 3.7** Line 349 – this line would benefit from further development or details on the "regional species succession reported by Aronson and Precht (2001)"; enough so that the reader does not need to go to the cited paper to get a clearer sense of what it is implied.

***Reply 3.7:*** *We modified the text in the manuscript to include this recommendation ( lines 352-355)*

**Comment 3.8** Line 371-372 - Please clarify - Not clear if the contrast with Valles et al refers to identifying ecological patterns within reef types (rather than among reef types) or by using geomorphic zones or metrics other than coral coverage or a combination of any of these.

***Reply 3.8:*** *We modified the text in the manuscript to clarify the contrast: (lines 383-387)*

Comment 3.9 Line 376 – missing bracket.

***Reply 3.9:*** *Done*

**Comment 4.0** Line 397-399- I was not sure about the basis for this statement about recruitment failure –it would benefit from further development or clarification in the Discussion or in the Conclusion itself.

***Reply 4.0:*** *We modified the text in the manuscript to clarify our conclusions.*